# Gamma Radiation Sterilization Dose of Adult Males in Asian Tiger Mosquito Pupae

**DOI:** 10.3390/insects10040101

**Published:** 2019-04-08

**Authors:** Andre Ricardo Machi, Raquel Rodrigues Mayne, Márcio Adriani Gava, Paula Bergamin Arthur, Valter Arthur

**Affiliations:** 1Department of Irradiation Technology Center—CTR, University of São Paulo/Institute of Nuclear Energy Research—IPEN/USP, São Paulo SP 05508-000, Brazil; paula.bergamin@hotmail.com (P.B.A.); arthur@cena.usp.br (V.A.); 2Department of Radiobiology and Environment—University of São Paulo/Center for Nuclear Energy in Agriculture—CENA/USP, University of São Paulo, Piracicaba SP 13400-970, Brazil; raquel.mayne@hotmail.com; 3Análises e Estudos Biológicos-AEB—Sítio Santa Isabel, 95 km Road, Rio Claro-SP-Charqueada SP 13515-000, Brazil; marciogava@asrlaboratorio.com.br

**Keywords:** *Aedes albopictus*, ionizing radiation, pupae, mosquitoes, sterile

## Abstract

The pathogen-carrying tiger mosquito, *Aedes albopictus*, has spread from the Western Pacific and Southeast Asia to Europe, Africa, the Middle East, North and South America, and the Caribbean. This species of mosquito transmits arboviral infections, such as yellow fever, chikungunya, dengue, zika, and several encephalitides. The objective of this research was to provide a radiation dose inducing sterilization in adult male *Ae. albopictus* in the pupal stage. A cobalt-60 source of gamma radiation at a dose rate of 381 Gy/h was used. The pupae were irradiated with doses of 0 (control), 20, 30, 40, 50, and 60 Gy. Each treatment had a total of five replications using 60 pupae. After irradiation, the different phases of *Ae. albopictus* development (egg, larva, pupa, and adult) in the F1 generation were observed daily. Parameters such as viability, fertility, longevity, and mortality were recorded. The results from these studies showed that a dose of 60 Gy was necessary to sterilize 100% of the male *Ae. albopictus* pupae.

## 1. Introduction

The invasive nature of the Asian tiger mosquito, *Aedes albopictus*, was described in Albania in 1979 [1]; subsequently, the species was introduced into the United States. Since then, several introgressions of Asian tiger mosquito across the world have been reported. Prior to 1980 [2], the species was also reported in Belgium, France, Italy, Switzerland, the Middle East, Cameroon, Equatorial Guinea, and Nigeria. In the New World, *Ae. albopictus* spread to Mexico by the early 1990s [3,4], and after this [5], all countries of Central America confirmed its presence in 2010.

In Italy, the Asian tiger mosquito has become the most serious pest mosquito species [6,7] as it is a potential vector of arboviruses such as yellow fever, dengue, chikungunya, zika, and other infections, such as *Dirofilaria immitis*, which causes heartworm parasitism. However, the introgressions suggest diverse geographic origins for this invasive mosquito species. In the New World, Costa Rica and Panamá have the same groups of population origins as in Latin America [8], whereas Hawaii, Cameroon, USA, and Italian populations belong to other groups and do not share haplotypes with the Brazil populations, indicating a separate introduction of this strain into Brazil.

*Ae. albopictus* was first detected in Brazil (São Paulo State) in 1986 and is currently in 24 of the 27 Brazilian Federal Units [9,10]. Although the species has not yet been cited as a main dengue vector in Brazil, studies under artificial conditions suggest that this species can be infected with and subsequently transmit the dengue virus [11]. The laboratory-based Brazilian populations of *Ae. albopictus* have shown high vector competence for the chikungunya virus, reinforcing the importance of the entomological and epidemiological surveillance of this mosquito species in the country. In tropical zones, *Ae. albopictus* has developed a photoperiodic egg diapause and freezing tolerance [12,13,14]. The occurrence of this species is now widespread in the USA, Central America, South America, and is currently established all over Africa [15].

Several studies were designed to develop a vaccine against these arboviruses [16,17]; however, the current means of control is the elimination of the larval phase in local areas by preventing standing water as well as chemical control. These are the most frequently used methods of vector control [18].

Consequently, the intense use of these compounds by both governmental campaigns and citizens (i.e., constant and uncontrolled household self-application) has selected resistant populations to nearly all classes of insecticides available [19]. Compounds such as organochlorines, carbamates, organophosphates, and pyrethroids have been successively administered since the 1950s to control mosquito populations [20], although no knockdown resistance (kdr) insecticide resistance mutations have been found for *Ae. albopictus* in Latin America [21,22].

Conventional methods do not produce satisfactory results; hence, to preserve the environment, new techniques must be applied. One such method is the sterile insect technique (SIT); this method involves the release of massive numbers of sterile males that compete with fertile wild males for mating opportunities with wild female insects [9,10]. The first release of adult mosquitoes irradiated in the pupae stage was performed in 1959–1960 in Florida to control *Anopheles quadrimaculatus.* However, the result was not satisfactory due to the low quality of sterilization [23,24]. In 1960–1961 in Pensacola, Florida, high doses of 110–180 Gy were applied in the pupal stage, which reduced the competitiveness of the adults, with irradiation causing the ineffectiveness of the release.

*Culex quinquefasciatus* irradiated at 60–120 Gy was effectively released in India and Florida between 1967 and 1974 [25]. These studies confirmed that an irradiated ion during the first hours (0–24 h) of the pupal stage can be more detrimental in this stage, in comparison to older pupae, where somatic damage was lower. When older pupae were irradiated, their competitiveness was not affected. The chemosterilizant thiotepa in water was used in a project targeted against *Cx. quinquefasciatus*, *Ae. aegypti*, and *Anopheles stephensi* pupae, funded by the World Health Organization (WHO) and by the United States Public Health Service (USPHS) in cooperation with the Indian Council of Medical Research (ICMR). The mass rearing facility in New Delhi produced up to one million males per day at a cost of USD $40 per million. The sterilization induced 99% sterility [26,27].

In Mombasa, Kenya, translocation homozygote strains were introduced to replace the wild *Ae. aegypti* population; however, fitness studies conducted in the field and laboratory showed that the translocation strains had low fertility, rate of larval development, larval and adult survival, and low mating competitiveness; consequently, none of the translocation strains became established in the field, and they rapidly disappeared [28].

In 2011, a transgenic lineage termed RIDL (release of insects carrying a dominant lethal genetic system) in *Ae. aegypti* strains [29] was liberated to suppress populations in Grand Cayman, which resulted in approximately 80% suppression of the wild *Ae. aegypti* population in the test area when compared with the experimental control area (76–78%). In Brazil, the release of transgenic insects reduced the *Ae. aegypti* population density by 85% in areas located in the Juazeiro and Jacobina municipalities (Bahia State) [30].

In Italy, from 2004 until currently, research has been conducted to investigate the feasibility of introducing the SIT strategy to control *Ae. albopictus.* In the first experiment, adults of male pupae showed significant results for a high sterilization rate in comparison with control treatments; research is continuing on a large scale to improve rearing efficiency [31,32].

The incompatible insect technique (IIT) uses the cytoplasmic incompatibility characteristics of *Wolbachia* to suppress a population. Although population replacement approaches can be performed with the simultaneous release of both male and female insects, population-suppression strategies should aim to release only males. Besides *Ae. aegypti*, other mosquito species, such as *Ae. polynesiensis* (South Pacific), *Ae. albopictus* (Italy), and *Culex pipiens quinquefasciatus* (Southwestern Indian Ocean), are also being tested in different countries to demonstrate the feasibility of using IIT for the population control of mosquito species [32].

The use of *Wolbachia* to reduce dengue transmission is being tested in several countries [33]. *Wolbachia*-infected mosquitoes are released in large numbers weekly, and the bacteria are transmitted to the wild *Ae. aegypti* population. Field trials have been conducted at four sites in Australia, one in Vietnam, and two in Indonesia; experiments in Brazil have been underway since September 2014 using *Ae. aegypti* infected with the wMel *Wolbachia* strain. Laboratory and field assays performed in Brazil showed that wMel infection has no detrimental effects on the Brazilian *Ae. aegypti* mosquito population. This technology can be used to control mosquito populations where local access is impossible (e.g., urban areas) [34,35,36]. Large-scale liberation has been conducted in Brazil (Niteroi and Rio de Janeiro), Colombia (Bello and Medellin), and Indonesia (Yogyakarta).

In Brazil, the SIT technique has been applied only to *Ae. aegypti* pupae since 2015 [37,38]. These previous studies showed that few mosquito sterilization programs in Brazil have been reported in the literature, and none have focused on *Ae. albopictus*. [23]. Thus, the success of SIT in vector control programs involves an area-wide approach combined with the release of large numbers of competitive sterile males in urban and rural areas with the highest risk of arbovirus transmission and highly diverse human populations [10,11,12]. Despite important advances in the development of molecular mechanisms for inducing male sterility [13], sterilization by irradiation remains the most practical method to sterilize mosquitoes in terms of cost and efficiency. Thus, the implementation of intensive control programs for *Ae. albopictus*, which is also a potential vector of human diseases, is necessary. This species is particularly suitable for the application of the sterile insect technique (SIT) due to its urban-related distribution, recent introduction, vector potential, low population density, which can be maintained by conventional control measures, and ease of mass rearing.

Therefore, *Ae. albopictus* is a potential candidate for SIT application. The objective of this research was to induce a sterilizing radiation dose for adult male *Ae. albopictus* irradiated in the pupal stage.

## 2. Materials and Methods

The research was performed in the Laboratory of Environment and Radiobiology, Center of Nuclear Energy in Agriculture (CENA/USP), at the Universidade de São Paulo, Piracicaba-São Paulo State, Brazil and Análise e Estudos Biológicos (AEB), Charqueda-São Paulo State, Brazil. The mosquitoes used in this experiment were obtained from the mass rearing colony at the Entomology Laboratory at LAFICAVE–IBEX–FIOCRUZ (Oswaldo Cruz Foundation), Benfica, RJ and had been maintained in the laboratory for more than three years.

### 2.1. Mosquito Source: Rearing and Preparation

Insects were reared under standard laboratory conditions (25 °C, 75% relative humidity (RH), 12 h photoperiod). Adults were kept in Plexiglas cages (50 × 50 × 50 cm). The cages were supplied with deionized water and 10% honey solution and contained adult males and females aged 5 days. During this period, the mosquitoes were fed with sheep blood via a vacuum collector. Females laid eggs in Petri dishes containing deionized water and a strip of white filter paper. After oviposition, the filter paper was removed from the cage and left to dry in a closed plastic container.

After four days, the eggs were counted and placed in a 1.0 L closed bottle with 750 mL of deionized water and 0.05 mg of food to stimulate hatching. Larvae were reared in white plastic trays (31 × 22 × 10 cm) at a density of 200 larvae/tray containing 2.5 L deionized water and provided with TetraMin fish food (0.20 mg; Tetra Werke, Melle, OS, Germany).

To separate males from females in the pupal stage, a device comprising two overlapping acrylic plates with an adjustable, downward-pointing, and wedge-shaped space was used, into which the pupae could be inserted. Pupae were separated on the basis of size by regulating the thickness and angle of the wedge-shaped space using four control knobs, one for each angle. The lower opening was adjusted so that the larger female pupae were retained in a layer in the tapering space between the acrylic plates. The smaller male pupae were drained into a plastic tray placed below. The operation was completed by opening the wedge and flushing the female pupae into a second receptacle (Model 5412, John W. Hock Company, Gainesville, FL, USA).

### 2.2. Irradiation Procedure

Each male pupa was transferred separately to a Petri dish containing white filter paper with a 10-mm layer of deionized water to sustain them. Afterward, they were placed in a Gammacell-220 irradiator (Atomic Energy of Canada, Ottawa, ON, Canada) located in then Center for Nuclear Energy in Agriculture—CENA/USP and exposed to cobalt-60 as a gamma radiation source at a dose rate of 381 Gy/h.

Doses in the irradiator were mapped by six Gammachrome dosimeters (Harwell Dosimeters, OX, UK) with a range of 0.1 to 3 kGy and read using a Genesy 20 spectrophotometer (Thermo Fisher Scientific, Waltham, MA, USA). Doses were certified by the Institute for Energy and Nuclear Research (IPEN), São Paulo, Brazil. The traceability of dose measurements was maintained by comparison with the international service assurance dose offered by the International Atomic Energy Agency, Vienna, Austria.

The Petri dishes were centered in the irradiator to ensure the uniformity of the radiation was not disrupted. The six dosimeters were positioned as follows: one on top of the dish, one at the bottom, and four equally spaced at lateral positions. The uncertainty in each dish was ±1.6%. The variation in measured doses was ±1.5% in the Gammacell-220 source.

Male pupae were irradiated at doses of 0 (control), 20, 30, 40, 50, and 60 Gy, with a total of five replications with 60 pupae per treatment.

After irradiation, the Petri dishes containing the adult males were transferred to new cages with 60 virgin females. The different phases of *Ae. albopictus* (egg, larva, pupa, and adult) in the F1 generation were evaluated daily for viability, fertility, longevity, cycle life duration of immature stages (duration stage), and mortality.

### 2.3. Statistical Analysis

A completely randomized experimental design was used in each experiment with five replications per treatment. Data were initially submitted to the Shapiro–Wilk normality test at a 5% level of significance. Proportional data were analyzed using a generalized linear model with binomial distribution. The graphics were produced with Microsoft Excel 2013 (Microsoft^®^, Redmond, WA, USA) and R software [39].

The data with multiple comparisons regarding the number of mosquitoes on average in the larva, pupa, or adult stage when submitted to the different doses of gamma radiation were compared using the *glht* function of the multicomp package in R. The survival rate of adult males from irradiated pupae and their longevity under various experimental conditions were used for Kaplan–Meier survival analysis.

## 3. Results

### 3.1. Radiation Effects in the F1 Generation from Irradiated Male Pupae

In the irradiation studies, the levels of damage were proportional to the increase in radiation doses compared with controls. This phenomenon was observed primarily in older pupae subjected to doses of 50 Gy. The dose of 60 Gy was administered to all ages, and the results were compared with other doses and the controls. Doses above 30 Gy caused deformations to appear in the larval body (personal observations). Higher doses of 40 or 50 Gy resulted in lower larvae and pupae numbers, whereas in doses of 60 Gy, no larval survival was observed. Thus, 60 Gy was the effective sterilizing dose.

The results for the cycle life in each stage (egg, larva, and pupa) in days showed that some irradiation treatments were overall slightly lower than in controls. However, at dose levels of 50 Gy, eggs developed to the larval L1-stage (L1). At doses of 40 and 50 Gy, however, the cycle duration time increased when there was an increase in age and dose, although no statistical difference was observed in comparison with the controls.

To rate the percentage of L1–L4 larvae that transformed in the pupal stage, gamma radiation effects were measured. Gamma radiation caused a significant reduction in pupae transformation: less than 60% transformed in the 20 Gy dose compared with 80% in the control. This difference was shown to be statistically significant (*p* < 0.001). At the 50 Gy dose, approximately 3% transformation was observed compared with the 60 Gy dose, where there was no observed L1–L4 larvae stage (Table 1 and Figure 1).

Figure 1a–d shows that almost 90% of the L1 larvae in the control dose transformed into pupae, whereas for the irradiated doses, 70% reached the pupal stage. However, we observed a gradual decrease in the survival of L1 larvae when there was an increase in dose that resulted in low pupa production (pupation), especially in doses from 30 Gy onward.

From doses at 50 Gy, no larvae were observed in stage L1 and the effects of gamma radiation on irradiated treatments from doses of 20 Gy onward were harmful, being statistically significant compared with the control treatment.

### 3.2. Gamma Radiation Effects on Longevity in the F1 Generation from Male Pupae

The radiation effects on pupal age were proportional to the increase in the doses that affected adult male viability. Younger pupae were less susceptible to radiation, while older pupae showed a decreased survival ability in all treatments.

In the first seven days, there was no observed decrease in the lifespan of pupae in any treatment group studied (Figure 2a–d). A slight decrease in longevity was observed at doses of 20 and 30 Gy compared with controls. However, the difference was not statistically significant. At the higher dose of 40 Gy, differences were statistically significant (*p* > 0.001). At the 60 Gy dose, the data obtained for male adult survival and male adult longevity were obtained from the P generation and compared with the other doses in the F1 generation. This was conducted because there were no pupae or adults in the F1 generation for this dose.

When comparing the viable percentage of adult males in Table 2, we observed a decrease proportional to the increased doses in all ages of irradiated treatments compared with the controls. We observed that the longevity of the males increased with the age of the pupae at the time of irradiation. From the initial doses of irradiation, a decreased percentage of viability was observed. At the dose of 20 Gy, viability was approximately 51% compared with 77% observed in the controls. At the 30 Gy dose, the percentage was significantly lower than 30% in all ages (*p* > 0.001). At 0–48 h at the 40, 50, and 60 Gy doses, the longevity of adult males in the P and F1 generations showed a significant reduction compared with the controls (*p* > 0.001).

## 4. Discussion

Overall, the radiation dose required to produce sterility was proportional to an increase in the dose, as expected. Some authors reported that increasing the radiation doses in pupae in mosquito species from the genus Aedes causes a large reduction in the number of male adults [40,41,42,43,44,45]. Our study supports those conclusions.

Here, we showed that sterility was reached at the 60 Gy dose for all pupae ages (0–48 h). Other authors used larger doses than those reported in the present study [23,41]. For example, irradiating *Ae. aegypti* male pupae aged >24 h resulted in a 99.9% sterility rate with a dose of 78 Gy [46]. In another study, exposing one-day-old male pupae produced a 100% sterility rate using 70 Gy [41]. The results of this study showed that irradiation reduced male survival, with a rate of approximately 99% of adults eradicated at a dose of 40 Gy. However, to reach a 100% sterilization rate, it was necessary to use a 50 Gy dose. Our results are similar to a study in Italy that induced sterilization in *Ae. albopictus* pupae with a 40 Gy dose; however, a 60 Gy dose was needed to achieve a 100% sterilization rate [42].

The adult emergence rate is considered to be an important parameter for measuring pupal irradiation. The radiation effects on insect pupae are greater than on adults, with the latter generally leading to a sterile male competing almost equally with wild males [47,48,49,50,51]. However, irradiation at the pupal stage is more convenient for practical reasons because the handling of pupae is easier than handling the more fragile adult mosquitoes.

A significant reduction in male adult longevity was observed in radiation treatments, especially at doses of 40 Gy or higher. A slight reduction was also observed in other treatments when compared with controls, although there was no significant difference among the doses.

Other studies also reported a significantly negative effect on male adult longevity after the pupal irradiation of *Ae. albopictus* at 40 Gy or higher [41,50]. This was likewise observed in other species, such as *Anopheles pharoensis* and *Anopheles stephensi* irradiated at more than 100 Gy [48,52].

The possible causes of somatic injuries induced by radiation treatment that result in decreased longevity have been previously described [51,52,53]. This negative effect of radiation treatment appeared to be stronger during the last days of the males’ lives. In the present study, it was approximately two weeks before such effects were observed. However, the results varied in some reports [49]. In these studies, *An. pharoensis* pupae (males and females) irradiated with low doses in the range of 4.8 to 68 Gy exhibited no reduction in longevity compared with controls (no irradiation).

Other studies on *Anopheles arabiensis* found that low doses, such as 4–15 Gy, could prolong the longevity of adult males [49]. The phenomenon of hormesis is proposed to explain the protective effect of a low radiation dose, which could stimulate defenses involved in cell repair mechanisms [53]. However, this was not the case in the current study.

The SIT methodology has not been attempted in Brazil, although studies of this nature have been conducted on several other mosquito species, including *Ae. aegypti*, which is a closely related species and may be taken as a useful example. Thus, the dose determined in this manuscript was used for the first time for the Brazilian strain of *Ae. albopictus*. Although positive, the results reported here need to be confirmed through a larger trial and with wild mosquitoes in the field, because laboratory strains of many generations could produce limited results in future pilot trials.

In future investigations, the main aims should be to evaluate the following: (1) the number of sterile males to be released, (2) a comparison of costs and benefits with conventional methods, (3) methods of improving the rearing efficiency by increasing larval density through introducing new larval diets and decreasing larvae numbers in plastic trays, (4) processes to increase pupal production by sieving at 30–36 h after the start of pupation instead of at 24 h, (5) the development of a larger greenhouse for colony maintenance, (6) testing for a lower sterilizing dose, (7) a pilot trial of population suppression and operational implementation in the field, (8) improvements in the quality of sterile males, and (9) the mass rearing of wild mosquitoes to study. 

## 5. Conclusions

The results presented here are important for mosquito species control programs that include SIT technology for efficient integrated pest management (IPM) in Brazil. From the results of longevity and gamma radiation doses, an effective sterilization dose was determined for male pupae of *Ae. albopictus* at 60 Gy.

## Figures and Tables

**Figure 1 insects-10-00101-f001:**
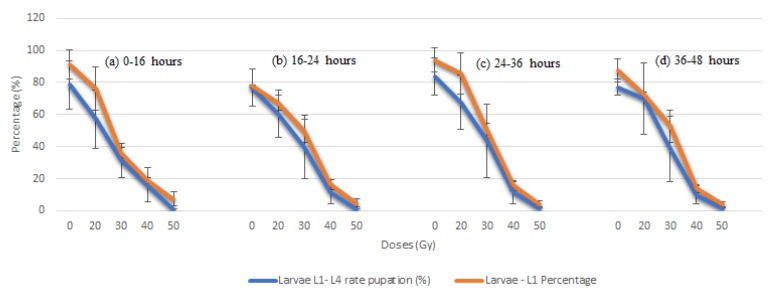
Percentage (±SD) of pupation of L1–L4 larvae from adults irradiated in the pupal phase at different ages: (**a**) 0–16 h, (**b**) 16–24 h, (**c**) 24–36 h, and (**d**) 36–48 h.

**Figure 2 insects-10-00101-f002:**
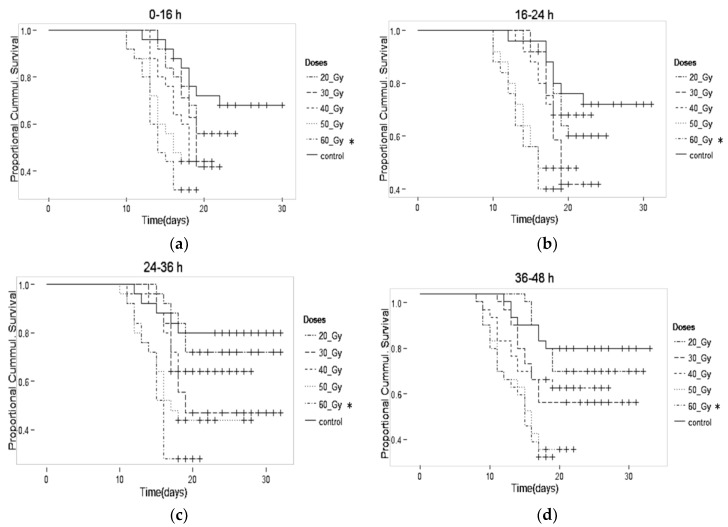
Adult male survival for pupae irradiated at different ages (**a**) 0–16, (**b**) 16–24, (**c**) 24–36, (**d**) 36–48 h. * Adult male at a dose of 60 Gy in (P) generation.

**Table 1 insects-10-00101-t001:** Total numbers of eggs, larvae and pupae, mean duration eggs, larvae and pupae stage, and larvae L1–L4 rates after pupae irradiation at different ages of *Ae. albopictus.*

^a^ Age (h)	Dose Level (Gy)	Total No. Eggs	Duration of Egg Stage ^b^ (days)	No. Total Larvae L1	Duration Larvae L1–L4 Stage	No. Total Pupae	Duration of Pupa Stage (days)	Larvae L1–L4 Rate (%) Transformed in Pupae
0–16	0	2975	3.1 ± 1.5	2712	5.8 ± 2.1	2493	3.1 ± 2.3	78.4 ± 6.7
	20	2750	2.9 ± 1.6	2099	5.6 ± 2.3	1575	2.8 ± 2.0	57.2 ± 6.0 *
	30	2.403	2.7± 1.5	2403	5.6 ± 2.1	872	312	2.9 ± 2.1
	40	1.599	2.9 ± 1.4	1599	5.8 ± 2.3	302	49	3.2 ± 1.9
	50	1.106	3.3 ± 1.3	1106	5.9 ± 2.1	182	14	3.2 ± 1.5
	60	1.000	0.0 ± 0.0	1000	0.0 ± 0.0	0	0	0.0 ± 0.0
16–24	0	3988	3.4 ± 2.3	3108	7.7 ± 3.4	2803	3.0 ± 2.8	77.0 ± 6.6
	20	3857	3.2 ± 2.2	2594	6.5 ± 3.5	1101	2.7 ± 2.3	60.4 ± 6.1
	30	3203	2.9 ± 2.4	1603	6.2 ± 3.2	203	2.7 ± 2.0	39.8 ± 4.1 *
	40	2498	3.6 ± 2.8	409	8.3 ± 3.3	34	3.1 ± 1.9	11.8 ± 2.3 *
	50	2099	3.5 ± 2.6	100	8.6 ± 3.2	19	3.3 ± 2.0	1.1 ± 0.1 *
	60	1902	0.0 ± 0.0	0	0.0 ± 0.0	0	0.0 ± 0.0	0.0 ± 0.0
24–36	0	4497	3.3 ± 3.1	4229	7.6 ± 3.8	3697	2.9 ± 3.4	83.7 + 7.3
	20	4518	3.3 ± 3.3	3876	6.3 ± 3.6	1103	2.6 ± 3.2	67.5 + 6.3
	30	4398	3.2 ± 3.4	2203	6.0 ± 3.4	129	2.8 ± 3.0	43.5 + 4.6 *
	40	2998	3.3 ± 3.1	469	8.7 ± 3.3	27	3.2 ± 2.5	11.4 ± 2.9 *
	50	2303	3.5 ± 3.0	98	9.0 ± 3.4	28	3.3 ± 2.4	1.5 ± 0.1 *
	60	2101	0.0 ± 0.0	0	0.0 ± 0.0	0	0.0 ± 0.0	0.0 ± 0.0
36–48	0	4336	3.3 ± 3.2	3789	7.7 ± 3.8	3687	2.9 ± 3.3	77.1 ± 6.4
	20	5125	3.2 ± 3.2	3699	6.5 ± 4.1	1201	2.8 ± 3.0	69.9 ± 6.0
	30	3011	3.1 ± 3.0	1590	6.4 ± 3.9	72	2.8 ± 2.3	38.7 ± 3.9 *
	40	2901	3.4 ± 3.2	399	9.1 ± 3.4	15	3.3 ± 1.2	9.9 ± 2.5 *
	50	2108	3.6 ± 2.9	82	9.3 ± 2.9	16	3.4 ± 1.6	1.6 ± 0.1 *
	60	1998	0.0 ± 0.0	0	0.0 ± 0.0	0	0.0 ± 0.0	0.0 ± 0.0

* Mean (±SD) in rows and columns were statistically significant (*p* > 0.001). ^a^ h—(hours). ^b^ Duration stage—duration life cycle history of each immature phase of *Aedes albopictus* in days.

**Table 2 insects-10-00101-t002:** Mean longevity of adult males and viable percentage of transformed pupae in adult males from irradiated pupae at various ages.

**0–16 h**	**Adult Male (%)**	**Male Mean Survival (days)**
Control	75.9 ± 3.5	22.4 ± 2.9
20 Gy	48.1 ± 2.9 *	19.0 ± 2.1
30 Gy	12.2 ± 1.8 *	18.9 ± 2.2
40 Gy	2.0 ± 1.0 *	17.4 ± 2.0 *
50 Gy	0.3 ± 0.3 *	15.6 ± 1.9 *
^a^ 60 Gy	0.0 ± 0.0 *	14.7 ± 1.7 *
**16–24 h**	**Adult Male (%)**	**Male Mean Survival (days)**
Control	76.0 ± 3.7	23.2 ± 3.2
20 Gy	50.8 ± 3.0 *	20.2 ± 2.4
30 Gy	22.0 ± 2.3 *	19.2 ± 2.3
40 Gy	0.9 ± 1.1 *	18.8 ± 2.3 *
50 Gy	0.3 ± 0.2 *	15.9 ± 2.1 *
^a^ 60 Gy	0.0 ± 0.0 *	15.0 ± 2.0 *
**24–36 h**	**Adult Male (%)**	**Male Mean Survival (days)**
Control	75.8 ± 3.2	24.5 ± 3.3
20 Gy	48.1 ± 2.7 *	23.7 ± 3.2
30 Gy	17.9 ± 2.0 *	21.3 ± 3.1
40 Gy	0.5 ± 0.4 *	20.3 ± 3.1 *
50 Gy	0.2 ± 0.1 *	17.4 ± 2.4 *
^a^ 60 Gy	0.0 ± 0.0 *	15.6 ± 2.1 *
**36–48 h**	**Adult Male (%)**	**Male Mean Survival (days)**
Control	74.8 ± 3.0	22.84 ± 3.1
20 Gy	41.0 ± 2.3 *	22.68 ± 3.1
30 Gy	9.9 ± 1.6 *	19.5 ± 3.0
40 Gy	0.3 ± 0.2 *	17.8 ± 2.7 *
50 Gy	0.2 ± 0.1 *	14.1 ± 2.1 *
^a^ 60 Gy	0.0 ± 0.0 *	13.4 ± 1.8 *

* Means (±SD) in rows and columns were statistically significant the level of *p* > 0.001. ^a^ Adult male in the dose of 60 Gy is as the (P) generation.

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
