# Peer review of "Gamma Radiation Sterilization Dose of Adult Males in Asian Tiger Mosquito Pupae"

_insects, 2019, doi:10.3390/insects10040101_

Round 1

Reviewer 1 Report

This manuscript presents a study to determine the sterilising dose of gamma radiation in male Aedes albopictus, and to measure the emergence, fertility and longevity of resulting adults.  However, it is very difficult to tell whether this study was performed well, analysed suitably, or the results support the conclusions drawn since the manuscript is very difficult to follow and missing in crucial detail and clarity.  It would benefit greatly from revision by a native English speaker.  Furthermore, I am not convinced of the novelty or general interest of the results presented. This is not the first time that the sterilising dose has been published for the species, and so if the study is published I would like to see, at the very least, more in the Introduction and/or Discussion to describe the novelty and significance of this study in a wider context - for example, is there reason to think that the dose response would be different in these local strains compared to previously published dose response data in these species? Are there any observations in the current study which contradict the conclusions of previous studies, or which fill in gaps in the current knowledge?

Specific Comments:

The Introduction should be re-ordered and re-focussed, putting this study in the context of a pilot study of the SIT in Brazil, referring to similar studies and explaining why this work is important in a Brazilian setting.  As written, it doesn’t flow or tell a coherent story.

The method for irradiating pupae, the most important step in the study, needs to be described in much more detail – how were the pupae arranged in the irradiator? Were the replicates irradiated separately or at the same time?  What was the emitting dose rate of the irradiator at the time of the study? Do lines 92-95 describe the dose mapping preparation, or the irradiation of the pupae itself?

The methodology of the life-history trait studies need to be described in detail, particularly explaining what parameters were measured and how the data was analysed, as this is often not clear in the Results section.

Table 1 – Label column 1, is this pupal age at irradiation? Column 3 – is this average number of eggs per female, and if so why is it so low? Column 4 – what is ‘duration egg stage’, and why was it measured? Column 5 – how is total larvae defined, why does it vary so much, and how does the number of larvae relate to number of eggs?

Figure 1 – is this really the longevity of the progeny (male/female/total?) of the irradiated pupae? If so, why was this measured and not the longevity of the sterilised males, which is an important parameter? Is Table 2 the longevity of the irradiated pupae? It’s not clear. 

The order of the Results should reflect the study to make it clearer to follow – data relating to the irradiated males first, and then the data relating to their progeny.

The Discussion is not very informative, and just makes statements from previous studies without relating them to the current results. Only sterilisation is discussed, and not the other parameters measured in the P and F1 generations. No context is given for this study.  I suggest briefly discussing the sterilising doses first, comparing results to previous studies in these Aedes species and discussing and explaining any differences, and then going on to discuss the balance between sterilising dose and impact on performance, and finally say that the next steps towards determining the potential of the SIT for Aedes control in Brazil are stepwise evaluation of the performance of irradiated males and eventually pilot population suppression trials and operational implementation.  Give a recommendation for which dose to use, considering both sterility and longevity of irradiated males. 

Author Response

Thanks a lot for comments of our manuscript!

The novelty and significance are showed in our manuscript now, the main are: In Brazil, the SIT technique has been applied only to A. aegypti and never to A. albopicuts, besides no exist none attempt also.

 In Brazil, this specie is competitive as A. aegypti by niches and is in evolution and growth in country and the more important, occur during the whole year without pause, as in colder countries, where in only one period the mosquito is actively present, thus the favoring the maintenance of the virus circulating in the environment as dengue for example together with A. aegypti.

Introduction

The observations in the current study, conclusions on the previous studies and gaps. 

Answer: are showed in introduction (lines 36-115) and discussion (lines 398-445)

The Introduction should be re-ordered and re-focussed, putting this study in the context of a pilot study of the SIT in Brazil, referring to similar studies and explaining why this work is important in a Brazilian setting.  As written, it doesn’t flow or tell a coherent story.

Answer: We made now, the answers are in introduction! (lines 36-115)

Methods

The method for irradiating pupae, the most important step in the study, needs to be described in much more detail – how were the pupae arranged in the irradiator? 

Answer: each replicate by dose was irradiated. (Lines 188 – 206). 

Were the replicates irradiated separately or at the same time?  Answer: separately (lines 188-189)

What was the emitting dose rate of the irradiator at the time of the study? 

Answer: This already was in previous correction, is in the (line 191) 

Do lines 92-95 describe the dose mapping preparation, or the irradiation of the pupae itself?

Answer: Yes describe, are in (lines 189-200)!

The answers for all question in methods are in the lines 157 -235

Results

Table 1 – Label column 1, is this pupal age at irradiation? Answer: Yes!

Column 3 – is this average number of eggs per female, and if so why is it so low? 

Answer: No is average, But the total number of eggs per female

 Column 4 – what is ‘duration egg stage’, and why was it measured? 

Answer: It is the average time in days for larvae to hatch from stage L1. Duration is important to measure if gamma radiation affects the development of a mosquito and immature stage can be cause malformations in body of larvae for example.

Column 5 – how is total larvae defined, why does it vary so much, and how does the number of larvae relate to number of eggs? 

Answer: They are defined according to the total count of how many larvae of the first stage L1 hatch from eggs per treatment. In nature there is a low mortality from one phase to another of the mosquito naturally. However, when we irradiate the insects, the mortality from one phase to another increases significantly according to the increase of the dose of radiation used.

Figure 1 – is this really the longevity of the progeny (male/female/total?) of the irradiated pupae? If so, why was this measured and not the longevity of the sterilised males, which is an important parameter? Is Table 2 the longevity of the irradiated pupae? It’s not clear.

Answer: Only the adult male from irradiate pupae, longevity is in total and for days to survival, while that the columm aduls male percentage showed the survival 

Answer: Figure 1 shows the survival curve x mortality of male adults at different ages from irradiated pupae.

Answer: The table 2 show longevity of male adults sterilized

The order of the Results should reflect the study to make it clearer to follow – data relating to the irradiated males first, and then the data relating to their progeny.

Answer: The work was somewhat confusing, but the manuscript was thus, table 1 represents the data of pupae after irradiation, whereas figure 1 and table 2 represent their progeny.

Discussion

first, comparing results to previous studies in these Aedes species and discussing and explaining any differences, and then going on to discuss the balance between sterilising dose and impact on performance, and finally say that the next steps towards determining the potential of the SIT for Aedes control in Brazil are stepwise evaluation of the performance of irradiated males and eventually pilot population suppression trials and operational implementation.  Give a recommendation for which dose to use, considering both sterility and longevity of irradiated males. 

Answer: We made this correction in discussions section, thanks for this observation, (lines 398-445)

Thanks a lot

Best regards

Andre Machi

Reviewer 2 Report

Machi et al describe the effects of radiation on Asian tiger mosquito pupae arriving to the conclusions that 60Gy intensity significantly reduces adult male fertility. The paper show good quality data and I enjoyed reading it, however there are several issues when showing that data and explaining it.

Methods

The authors showed good control of radiation technique and measurement showing less than 1.5% variability. The numbers of samples and the replication times were also sufficient.

Results

The results are interesting; however, I would like to suggest that Table 1 be presented in a graphic format, with columns and error (SD) bars (except for the no. of total eggs) and showing percentage of larvae L1, percentage of pupae. Similar to what was done in Figure 1. What do the first set of numbers (0-16, 16-24, 24-36, 36-48) mean? Are those the time points after hatching at which radiation was administered? Authors should explain them probably in the methods section. Figure 1 misses standard deviations bars.

Statistics

Having a p value=0.001 is very unlikely. Usually values are smaller or bigger. All the p values in the paper are the same, 0001. I do not argue the significant of the value, but rather if it is presented properly. Please explain which test was used and provide the proper p value, either giving the exact p-value (i.e. p=0.008345) or the approximation expressed correctly (p>0.001). Please include SD instead of SE.

Discussion

The authors summarised findings from other research papers however they should mention in which state those mosquitoes were. For example, reference [26], which is a very similar study, the mosquitoes tested were adults, as opposed to larvae. The authors need to leave that clear.

Conclusion

The sentences need to be switched around, the conclusion is that the effective dose for sterilizing male pupae is 60 Gy. The other sentence denotes the significance of the study. Also in the conclusion they should mention the downside effects of this technique as well as other implication of radiation. They should mention the correlation of a higher radiation doses with lower sterility but with also higher fitness cost.

Style, formatting and grammar

The authors need to address significant grammar and style issues. Some sections of the paper do not make much sense. Please have some English editorial advice. The paragraphs need better formatting. It seems that every sentence Is its own paragraph. All the species names and abbreviations should be in italics, and consistent, either A. albopictus or Ae. albopictus, but not both across the paper. Figure and table labelling are incomplete. This is a good area where to write your number of samples (n) and the rest of the statistics (test done and p-value).

Author Response

Thanks a lot for your important comments under our manuscript

The results are interesting; however, I would like to suggest that Table 1 be presented in a graphic format, with columns and error (SD) bars (except for the no. of total eggs) and showing percentage of larvae L1, percentage of pupae. Similar to what was done in Figure 1. What do the first set of numbers (0-16, 16-24, 24-36, 36-48) mean?

Answer: I´m sorry for confuse, the numbers are the ages in which pupae were irradiated. We made graphic, thanks for this observation, is very good, the graphic is in (lines 295-299). We also are observed all data again, already that add (error SD) and modify some data showed in both tables of manuscript.

Are those the time points after hatching at which radiation was administered? 

Answer:  Is the average time of development of per dose by each immature phase (eggs, larvae and pupae). 

Authors should explain them probably in the methods section. Figure 1 misses standard deviations bars.

Answer: The deviation standard was not placed in this figure because we did a log rank analysis.

Statistics

Having a p value=0.001 is very unlikely. Usually values are smaller or bigger. All the p values in the paper are the same, 0001. I do not argue the significant of the value, but rather if it is presented properly. Please explain which test was used and provide the proper p value, either giving the exact p-value (i.e. p=0.008345) or the approximation expressed correctly (p>0.001). Please include SD instead of SE. 

Answer: I ‘sorry, was a write error, ok we leave in SD and also placed the statistics in manuscript that has desappear.

Discussion

The authors summarised findings from other research papers however they should mention in which state those mosquitoes were. For example, reference [26], which is a very similar study, the mosquitoes tested were adults, as opposed to larvae. The authors need to leave that clear.

 Answer: thanks for observation, we corrected this, but the reference [26] that now is reference [32] was made with pupae same.

Conclusion

The sentences need to be switched around, the conclusion is that the effective dose for sterilizing male pupae is 60 Gy. The other sentence denotes the significance of the study. Also in the conclusion they should mention the downside effects of this technique as well as other implication of radiation. They should mention the correlation of a higher radiation doses with lower sterility but with also higher fitness cost.

Answer: Ok made in Conclusion (lines 495-499)

Reviewer 3 Report

The manuscript "Gamma radiation on tiger mosquito pupae proposed as a Sterile Insect Technique (SIT) in Brazil" is a manuscript that is difficult to judge given the quality of the english language, For example, just starting by the current title does not express clearly something that guessing might be "Gamma radiation of Asian tiger mosquito pupae as a potential Sterile Insect Technique (SIT) for use in Brazil", which might be like the ultimate outcome from the study goal presented in lines 66-67.

That being said one major issue that i see with the study design was the use of a reference colony which has been established for over three years, and for which i did not see any attempt to discuss the limitations drawn from inferences based on mosquitoes that have been so many generations in the lab to make an extrapolation to the field.

The second major issue that i see is the lack of clear methods description. For example, no detailed description of the device, and the principple to separate pupae by sex, is presented, which is fundamental, for example, to replicate what this study did (see lines 81-82).

A third issue is the lack of references to previous work. For example, in lines 28-32 several commensta re made about the biology of the Asian tiger mosquito (bTW saying that Ae. albopictus common name is tiger mosquito is incorrect. See the common names database of the Entomological society of America https://www.entsoc.org/common-names?title=asian+tiger+mosquito&field_scientific_name_value=&tid=&tid_1=&tid_2=&tid_3=&tid_4=)

Author Response

Thanks a lot for important comments under our manuscript

That being said one major issue that i see with the study design was the use of a reference colony which has been established for over three years, and for which i did not see any attempt to discuss the limitations drawn from inferences based on mosquitoes that have been so many generations in the lab to make an extrapolation to the field. 

Answer: We made in discussions, (lines 435-445)!

The second major issue that i see is the lack of clear methods description. For example, no detailed description of the device, and the principple to separate pupae by sex, is presented, which is fundamental, for example, to replicate what this study did (see lines 81-82). 

Answer: Corrected in Material and Methods, thanks! (lines 173-180)

A third issue is the lack of references to previous work. For example, in lines 28-32 several commensta re made about the biology of the Asian tiger mosquito (bTW saying that Ae. albopictus common name is tiger mosquito is incorrect. See the common names database of the Entomological society of America https://www.entsoc.org/common-names?title=asian+tiger+mosquito&field_scientific_name_value=&tid=&tid_1=&tid_2=&tid_3=&tid_4=) Answer: corrected, very thanks for this observation!

Round 2

Reviewer 1 Report

The authors have made some effort to improve this manuscript based on reviewers comments, and the flow of the Introduction is somewhat improved.  However, the list of previous mosquito control efforts in the Introduction is not useful, without discussion or context as to how they relate to the present study.  The stated objective of the study has been narrowed to determining the sterilising dose, which more accurately reflects the scope of the study, and the title is now more accurate.

However, not all comments have been addressed – where I ask a question about the study it is not for my own benefit but the readers’, and it is a suggestion to clarify the point in the manuscript!  The author has commented on my observations about Table 1 but has not changed the table or legend, and these points still need clarification.  The measured parameters should be clearly defined in the Methods section.  The author has answered my other questions about the results section in the response, but not made the corresponding clarifications in the manuscript itself.  The text in the Results section is somewhat improved, and now at least describes the results obtained, even if a clear story is still not really presented.

I am not convinced that a confirmation of the sterilising dose in Ae. albopictus, which has been published previously, without discussion of why this dose might have been different to the published value, is a novel result. The Conclusion merely states that the sterilising dose has been established for Ae. albopictus, but this has been done already.  And a thorough editing for English is still required.

Author Response

The authors have made some effort to improve this manuscript based on reviewers comments, and the flow of the Introduction is somewhat improved.  However, the list of previous mosquito control efforts in the Introduction is not useful, without discussion or context as to how they relate to the present study.  The stated objective of the study has been narrowed to determining the sterilising dose, which more accurately reflects the scope of the study, and the title is now more accurate.

Answer: We made new modification seeking improve these points in lines we contextualize the introduction previous efforts control in lines (63-65 and 113 -126). Please if our corrections still not good, please can you help us to clarify more so that we can improve the manuscript.

However, not all comments have been addressed – where I ask a question about the study it is not for my own benefit but the readers’, and it is a suggestion to clarify the point in the manuscript!  The author has commented on my observations about Table 1 but has not changed the table or legend, and these points still need clarification.  The measured parameters should be clearly defined in the Methods section.  The author has answered my other questions about the results section in the response, but not made the corresponding clarifications in the manuscript itself.  The text in the Results section is somewhat improved, and now at least describes the results obtained, even if a clear story is still not really presented.

Answer: Dear editor, maybe we no understand your considerations previous, however we know that your questions it is no for your benefit and also thank you for your valuable considerations in our manuscript.

In table 1 we made new modifications in the legend and title of table 1 as requested, the changes are in blue color and deletions in red color, however we are no understood what others modifications must be made in table 1, I’m sorry. Please, if we no made still these modifications requested by you, please you could be clarifying for that we improve the manuscript?

The methods section defined in previous comments the measured parameters were placed in manuscript, are in the lines 169-172. if our corrections are wrong please, you can clarify! 

I am not convinced that a confirmation of the sterilising dose in Ae. albopictus, which has been published previously, without discussion of why this dose might have been different to the published value, is a novel result. The Conclusion merely states that the sterilising dose has been established for Ae. albopictus, but this has been done already.  And a thorough editing for English is still required.

The novelty is that SIT technique had been applied in Brazil only in A. aegypti and never made in A. albopictus previously.

The fact of similar dose had been found in our manuscript, are according to results of sterilizing doses found to dipteran families in overall, these doses vary between 20-160 Gy and to Aedes spp between 30 -60 Gy in overall.

And besides, this dose is the first found to a Brazilian strain of A. albopictus. Besides, this Brazilian strain is haplotypically different of strain populations of other countries as Italy, USA for example, we added this information in lines: 29-40. This mean also that the gamma radiation effects achieved with this dose applied in this strain Brazilian is no different in comparison to Italy country for example despite of geographic distance. However, can be different in comparison to other countries or others strain, we can added this information in manuscript on different sterilization doses with different strains, if necessary.

The previous answer in Reviewer 1 – round 1 on table1 are below, We modify the title and legend, however we are no understood what others modifications must be made in table 1, I’m sorry. Please, if we no made still these modifications requested by you, please you could be clarifying for that we improve the manuscript?

The previous answer to table 1 were these and added more information in blue color!

Table 1 – Label column 1, is this pupal age at irradiation? Answer: Yes! Now we made a update placed Age (h) in this column.

Column 3 – is this average number of eggs per female, and if so why is it so low? 

Answer: No is average, But the total number of eggs per female.

 Column 4 – what is ‘duration egg stage’, and why was it measured? 

Answer: It is the average time in days for larvae to hatch from stage L1. Duration is important to measure if gamma radiation affects development of a mosquito and immature stage can be cause malformations in body of larvae for example. Duration stage show an evolution history by life stage, for identify in which stage and dose, the gamma radiation effects were more stronger in this immature stage.

Column 5 – how is total larvae defined, why does it vary so much, and how does the number of larvae relate to number of eggs? 

how is total larvae defined: Answer: They are defined according to the total count of and how does the number of larvae relate to number of eggs: The larvae of the first stage L1 hatch from eggs per treatment.

why does it vary so much:  In nature there is a low mortality from one phase to another phase of the mosquito naturally. However, when we irradiate, the mortality from one phase to another phase increases significantly according to  increase of  dose used.

Thanks a lot

Reviewer 3 Report

The manuscript "Gamma radiation sterilization dose of adult male in 2 Asian tiger mosquito pupae" has greatly improved following the first round of reviews. 

Minor Comments

1) The manuscript english still needs to be carefully reviewed, but the journal editorial office probably can take care of that

2) In line 28 "The infectious parameters of Aedes albopictus were" needs to be changed to "The invasive nature of Aedes albopictus was"

3) In line 32 after "America", please cite the fact that Ae. albopictus has had several introgressions in Latin America as suggested by haplotype diversity, which suggest diverse geographic origins for this invasive mosquito species in the New World. See and Cite:

Futami K, Valderrama A, Baldi M, Minakawa N, Marin Rodríguez R & Chaves LF. 2015. New and common haplotypes shape genetic diversity in Asian tiger mosquito populations from Costa Rica and Panamá. Journal of Economic Entomology 108(2): 761-768.

Available at:

 https://doi.org/10.1093/jee/tou028

4) In line 55 it is important to refer to the fact that no kdr insecticide resistance mutations have been found for Ae. albopictus in Latin America. See and Cite:

Chaves LF, Kawashima E, Futami K, Minakawa N, Marin Rodríguez R. 2015.  Lack of kdr mutations in Asian tiger mosquitoes from Costa Rica. Bulletin of Insectology 68(1): 61-63.

Available at: 

Valderrama A, Chaves L, Futami K. 2016. Evaluación de mutaciones kdr en Aedes albopictus (Skuse) en Panamá (Datos preliminares). Revista Médica de Panamá. 36(2):30-32.

Available at:

http://www.revistamedica.org/index.php/rmdp/article/view/412

5) Please, use through the manuscript standard abbreviations for Culicidae genera. Especifically Cx. for Culex and Ae. for Aedes  Related to this point "C. pipiens quinquefasciatus" is not a valid species but "Cx. quinquefasciatus" so adjust the text accordingly.

Author Response

Thanks a lot for important comments under our manuscript

1) The manuscript english still needs to be carefully reviewed, but the journal editorial office probably can take care of that

Answer: Thanks for this observation!

2) In line 28 "The infectious parameters of Aedes albopictus were" needs to be changed to "The invasive nature of Aedes albopictus was"

Answer: We made, thanks

3) In line 32 after "America", please cite the fact that Ae. albopictus has had several introgressions in Latin America as suggested by haplotype diversity, which suggest diverse geographic origins for this invasive mosquito species in the New World. See and Cite:

Answer: We really like of these articles, thus added more information about these manuscripts in our manuscript. Lines: 28-51

Futami K, Valderrama A, Baldi M, Minakawa N, Marin Rodríguez R & Chaves LF. 2015. New and common haplotypes shape genetic diversity in Asian tiger mosquito populations from Costa Rica and Panamá. Journal of Economic Entomology 108(2): 761-768.

Available at:

 https://doi.org/10.1093/jee/tou028

4) In line 55 it is important to refer to the fact that no kdr insecticide resistance mutations have been found for Ae. albopictus in Latin America. See and Cite:

Chaves LF, Kawashima E, Futami K, Minakawa N, Marin Rodríguez R. 2015.  Lack of kdrmutations in Asian tiger mosquitoes from Costa Rica. Bulletin of Insectology 68(1): 61-63.

Available at: 

Valderrama A, Chaves L, Futami K. 2016. Evaluación de mutaciones kdr en Aedes albopictus(Skuse) en Panamá (Datos preliminares). Revista Médica de Panamá. 36(2):30-32.

Available at:

http://www.revistamedica.org/index.php/rmdp/article/view/412

Answer: we made the corrections. Lines: 60-61

5) Please, use through the manuscript standard abbreviations for Culicidae genera. Especifically Cx. for Culex and Ae. for Aedes Related to this point "C. pipiens quinquefasciatus" is not a valid species but "Cx. quinquefasciatus" so adjust the text accordingly.

Answer: Corrected!
